# Sea urchin-like microstructure pressure sensors with an ultra-broad range and high sensitivity

Xiu-man Wang[1], Lu-qi Tao[2], Min Yuan[1], Ze-ping Wang[1], Jiabing Yu [1], Dingli Xie[1], Feng Luo[1], Xianping Chen [1,2✉] & ChingPing Wong [3✉]

Sensitivity and pressure range are two significant parameters of pressure sensors. Existing pressure sensors have difficulty achieving both high sensitivity and a wide pressure range. Therefore, we propose a new pressure sensor with a ternary nanocomposite $Fe_2O_3/C@SnO_2$. The sea urchin-like $Fe_2O_3$ structure promotes signal transduction and protects $Fe_2O_3$ needles from mechanical breaking, while the acetylene carbon black improves the conductivity of $Fe_2O_3$. Moreover, one part of the $SnO_2$ nanoparticles adheres to the surfaces of $Fe_2O_3$ needles and forms $Fe_2O_3/SnO_2$ heterostructures, while its other part disperses into the carbon layer to form $SnO_2@C$ structure. Collectively, the synergistic effects of the three structures ($Fe_2O_3/C$, $Fe_2O_3/SnO_2$ and $SnO_2@C$) improves on the limited pressure response range of a single structure. The experimental results demonstrate that the $Fe_2O_3/C@SnO_2$ pressure sensor exhibits high sensitivity (680 $kPa^{-1}$), fast response (10 ms), broad range (up to 150 kPa), and good reproducibility (over 3500 cycles under a pressure of 110 kPa), implying that the new pressure sensor has wide application prospects especially in wearable electronic devices and health monitoring.

[1] Key Laboratory of Optoelectronic Technology & Systems, Education Ministry of China, Chongqing University and College of Optoelectronic Engineering, Chongqing University, Chongqing, China. [2] State Key Laboratory of Power Transmission Equipment & System Security and New Technology and School of Electrical Engineering, Chongqing University, Chongqing, China. [3] School of Materials Science and Engineering, Georgia Institute of Technology, Atlanta, GA, USA. ✉email: xianpingchen@cqu.edu.cn; cp.wong@mse.gatech.edu

In the past few years, pressure sensors are considered promising candidates for use in wearable devices, electronic skins, and human-machine interfaces due to their low cost, flexibility, simple fabrication process, high integration potential, among others[1–4]. In particular, pressure sensors are classified into four main types: capacitive[5–7], piezoresistive[8–10], piezoelectric[11–13], and triboelectric sensors[14–16]. Piezoresistive pressure sensors have multiple benefits, including low energy consumption, easy signal collection, simple device assembly, and high sensitivity. Recently, different microstructure or nanostructure geometries such as (interlocked microstructures[17], hollow-sphere microstructure[18], micropyramid array[19], and porous structure[20]) have been explored to improve the sensitivity of piezoresistive pressure sensors. Among them, the tapering geometry or spine structure confers a clever design that not only promotes signal transduction for high sensitivity, but also protects the bristle from mechanical breaking[21,22]. Similar structures have been used in mechanical sensors and have been shown to improve sensing performance. For example, Yin et al. reported that ZnO sea-urchin-shaped microparticles with a low-temperature solution process exhibited a high sensitivity of 121 kPa$^{-1}$ (pressure range 0–10 Pa)[23]. Lee et al. achieved a sensitivity of 2.46 kPa$^{-1}$ (pressure range 0–1 kPa) with a piezoresistive pressure sensor based on sea-urchin-shaped metal nanoparticles[24]. Furthermore, Shi et al. studied the urchin-like hollow carbon spheres, and reported that the sensitivity reached 260.3 kPa$^{-1}$ at 1 Pa[25]. Therefore, the piezoresistive pressure sensors have a high sensitivity, but only under a small pressure range. Additionally, without any additives, relying only on the structure and performance of conductor and semiconductor, it is difficult to achieve a high sensitivity and a wide pressure working range at the same time.

High sensitivity can be achieved with two conditions; low initial current and large output current changes under a certain pressure[26]. The conductivity of semiconductor is considerably low, so the initial current could be achieved at low level. In addition, the semiconductor/conductor interface piezoresistive effect is favorable for the change of contact area, which leads to a high output current change[26]. A depletion region and band bending occurs in the contact sections of the heterojunction, which induces the lower interfacial resistance and promotes the charge transport/transfer[27]. Heterojunctions have been used in many modern devices, including light emitting diodes (LEDs), photodetectors and solar cells[28–30]. Therefore, when the metal oxide semiconductor/C composite structure and the heterostructures are used in the pressure sensor, the sensing performance of the pressure sensor may be improved.

In this work, we propose a pressure sensor with nanostructure design of materials with contains metal oxide semiconductor/C composite structure and a heterostructure. When fabricated using the new nanostructure, the pressure sensor exhibited ultra-sensitivity and an ultra-broad-range. We chose Fe$_2$O$_3$ and SnO$_2$ as sensing materials, because of their low cost, environmental friendliness, and natural abundance. Sea-urchin-like Fe$_2$O$_3$ was synthesized through a hydrothermal method. This strategy involves the use of acetylene black carbon as a carrier, due to its strong conductivity and high specific surface. One part of acetylene black carbon encloses Fe$_2$O$_3$ particles, whereas the carbon materials part was embedded in the Fe$_2$O$_3$ needles gap, forming a Fe$_2$O$_3$/C structure. Furthermore, one part of the SnO$_2$ nanoparticles was dispersed into the carbon layer to form the SnO$_2$@C structures, whereas its other part adhered to the Fe$_2$O$_3$ needles surface to form the Fe$_2$O$_3$/SnO$_2$ heterostructure. Carbon improves the conductivity of a single metal oxide. Collectively, the synergistic effects of the three structures (Fe$_2$O$_3$/C, Fe$_2$O$_3$/SnO$_2$ and SnO$_2$@C) improved the limited pressure response range of a single structure. Notably, the Fe$_2$O$_3$/C@SnO$_2$ (3:1:4) pressure sensor exhibited a high sensitivity (680 kPa$^{-1}$), fast response (10 ms), broad range (up to 150 kPa) and good reproducibility (over 3500 cycles under a pressure of 110 kPa).

## Results

**Structural characterization**. The fabrication process used in this study is shown in Fig. 1. First, conductive materials were synthesized through a hydrothermal method. Then, a clean melamine sponge was soaked in the sample solution. After the electrode connection, a pressure sensor with a melamine sponge substrate was obtained. Figure 1b is an image of the pressure sensor.

The phase structures of acetylene carbon black, Fe$_2$O$_3$, SnO$_2$, Fe$_2$O$_3$/C (the mass ratio of Fe$_2$O$_3$/C 3:1), and Fe$_2$O$_3$/C@SnO$_2$ (the mass ratio of Fe$_2$O$_3$/C@SnO$_2$ 3:1:4) were characterized by X-ray diffraction (XRD), as shown in Fig. 2a. The acetylene carbon black revealed a broad peak at 20°–30°, corresponding to its (002) crystal plane. The XRD pattern with (012), (104), (110), (113), (024), (116), (214), and (300) was regarded as the formation of Fe$_2$O$_3$ (JCPDS 33-0664). The pristine SnO$_2$ nanoparticles exhibited a tetragonal structure (JCPDS 41-1445). Remarkably, the diffraction peaks of acetylene carbon black and Fe$_2$O$_3$ were observed in Fe$_2$O$_3$/C. Besides, as the carbon mass increased, the carbon peaks became stronger (Supplementary Fig. 1a). The XRD pattern of Fe$_2$O$_3$/C@SnO$_2$ exhibited broad peaks. All the diffraction peaks corresponded with single acetylene carbon black, Fe$_2$O$_3$ and SnO$_2$, indicating the Fe$_2$O$_3$/C@SnO$_2$ nanocomposite has a high purity. Notably, the peak of SnO$_2$ was stronger in the Fe$_2$O$_3$/C@SnO$_2$ (3:1:8) nanocomposites, implying a higher content of SnO$_2$ in this composite (Supplementary Fig. 1b).

The microstructures of the Fe$_2$O$_3$, Fe$_2$O$_3$/C (3:1), and Fe$_2$O$_3$/C@SnO$_2$ (3:1:4) were characterized by scanning electron microscopy (SEM), transmission electron microscopy (TEM), elemental mapping, and high-angle annular dark-field scanning transmission electron microscopy (HAADF-STEM) (Fig. 2b–i).

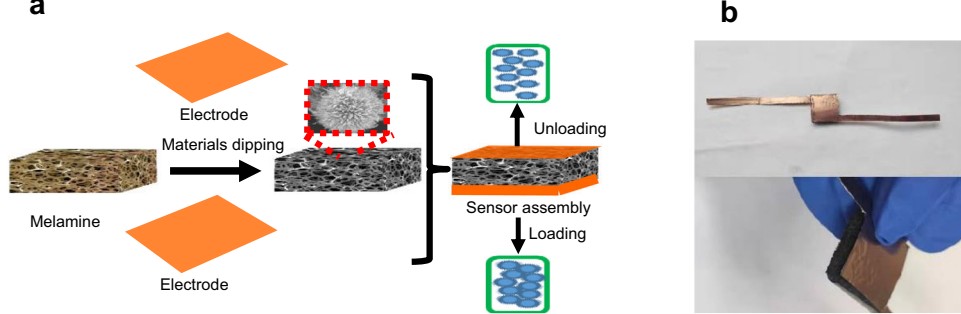

**Fig. 1 Preparation diagram and sensor images. a** Schematic illustration of the fabrication of the pressure sensor. **b** Images of the pressure sensor encapsulated with a copper tape.

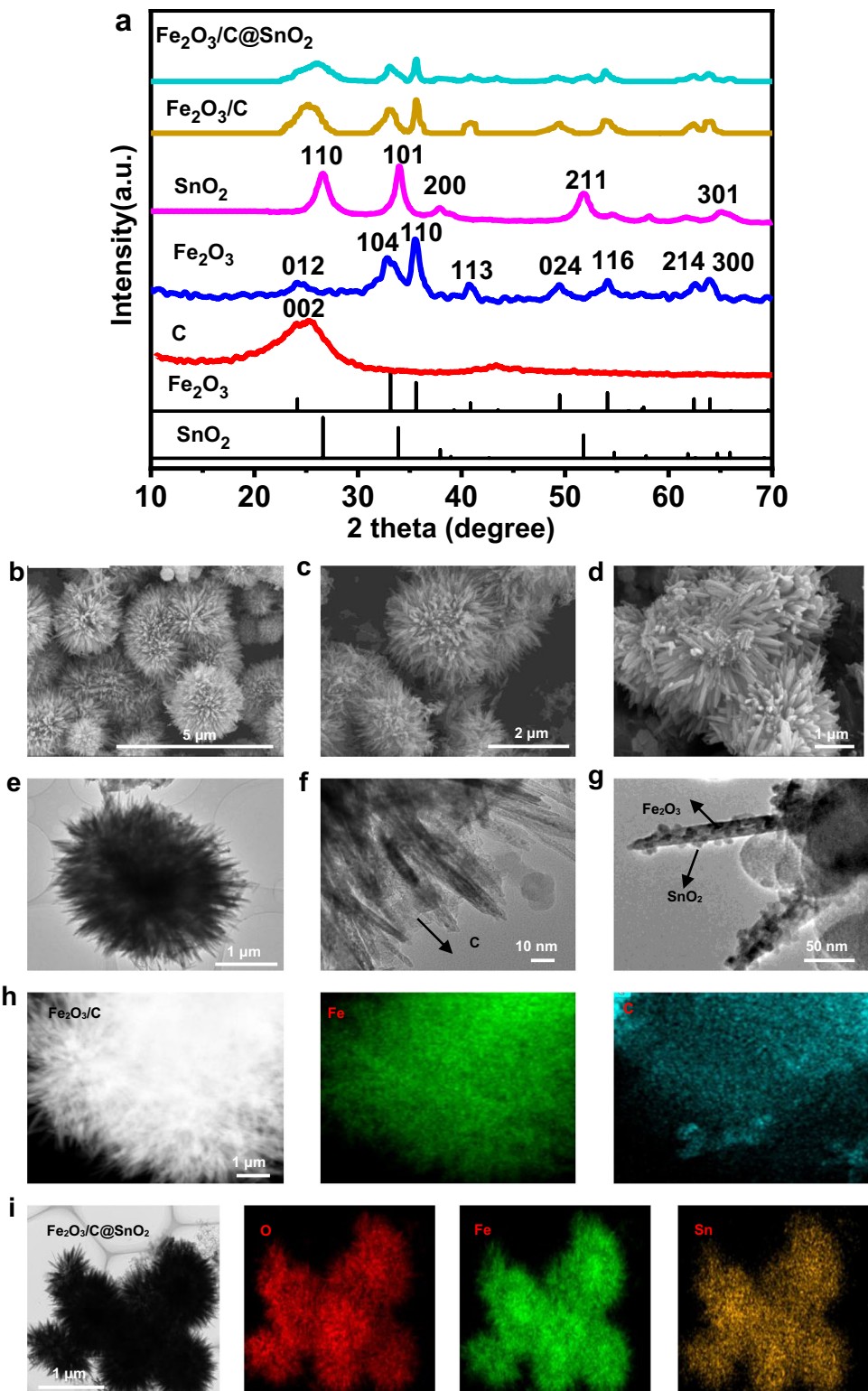

**Fig. 2 Fe₂O₃, SnO₂, Fe₂O₃/C (3:1), and Fe₂O₃/C@SnO₂ (3:1:4) microparticles structure and morphology. a** The XRD patterns of acetylene carbon black, Fe₂O₃, SnO₂, Fe₂O₃/C (3:1) and Fe₂O₃/C@SnO₂ (3:1:4). SEM images of **b** Fe₂O₃, **c** Fe₂O₃/C (3:1), and **d** Fe₂O₃/C@SnO₂ (3:1:4), TEM images of **e** Fe₂O₃, **f** Fe₂O₃/C (3:1), and **g** Fe₂O₃/C@SnO₂ (3:1:4), elemental mapping of **h** Fe₂O₃/C (3:1), and **i** Fe₂O₃/C@SnO₂(3:1:4).

The images of $Fe_2O_3$ reflect a typical sea-urchin-like structure, with a diameter of about 3 μm, as shown in Fig. 2b. Microstructures of the $Fe_2O_3/C$ nanocomposites are shown in Fig. 2c, f. In addition, one part of acetylene black carbon encloses $Fe_2O_3$ particles, while the other part of carbon materials was embedded in the gap of $Fe_2O_3$ needles, thereby forming a $Fe_2O_3/C$ structure. The SEM images of resulting the $Fe_2O_3/C@SnO_2$ (3:1:4), in which $SnO_2$ nanoparticles are visible, as shown in Fig. 2g. Moreover, one

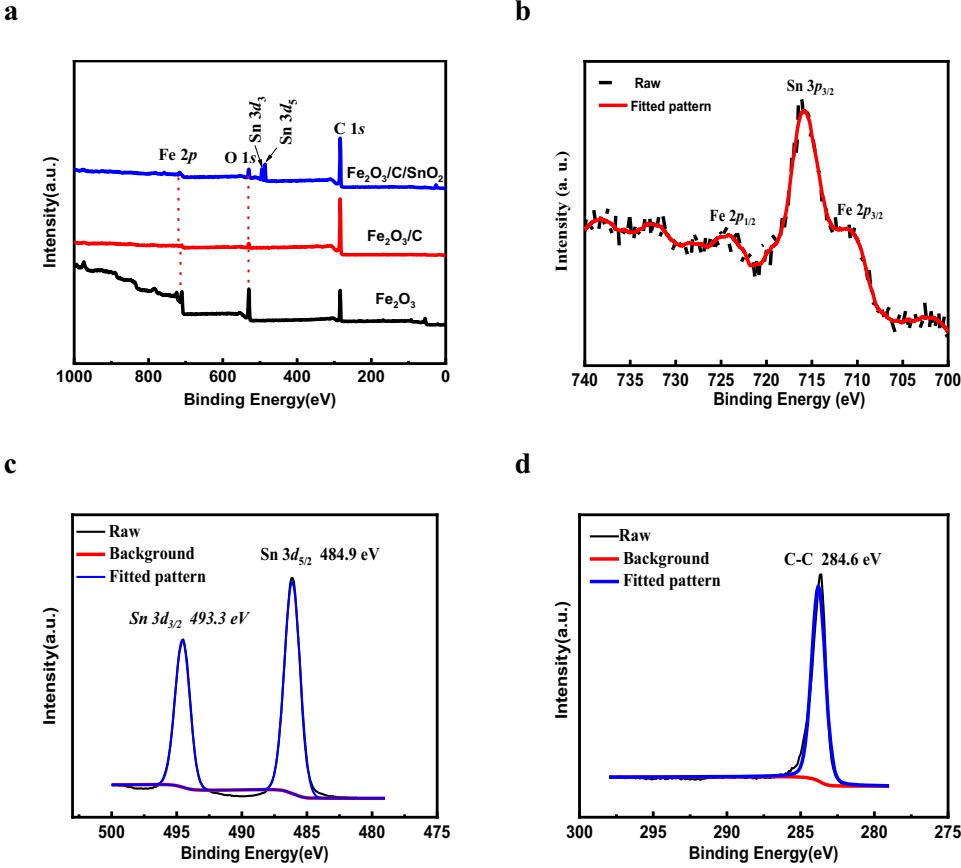

**Fig. 3 XPS spectrum. a** The XPS of $Fe_2O_3$, $Fe_2O_3$/C (the mass ratio of 3:1), $Fe_2O_3$/C@$SnO_2$ (the mass ratio of 3:1:4), **b** high-resolution curves of **c** Fe, Sn, and **d** C.

part of the $SnO_2$ nanoparticles was tightly attached to $Fe_2O_3$ needles, implying the formation of a heterojunction between $Fe_2O_3$ and $SnO_2$. Supplementary Fig. 2i shows the interfacial structure of the $Fe_2O_3$/C@$SnO_2$ (3:1:4), as revealed by a high-resolution TEM (HRTEM) technique. The $SnO_2$ nanoparticles grew on the (104) surface of the $Fe_2O_3$, along the direction of (110) $SnO_2$, forming a (104) $_{Fe_2O_3}$/ (110) $_{SnO_2}$ heterojunction. Moreover, the other part of $SnO_2$ nanoparticles penetrated the carbon layer, forming $SnO_2$@/C (Fig. 2g). It was noted that the $Fe_2O_3$/C@$SnO_2$ had three structures with $Fe_2O_3$/C, $SnO_2$@/C, and $Fe_2O_3$/$SnO_2$ heterojunction. As the mass of carbon and $SnO_2$ increased, the carbon layer and $SnO_2$ nanoparticles also became clear in the SEM images (Supplementary Fig. 2a-f), consistent with the XRD experimental results. Microstructures and compositional distribution of nanocrystals were further determined by STEM and EDX mapping, respectively. The tin signal for $SnO_2$, iron signals for $Fe_2O_3$, and carbon signals were overlapped completely across the entire sample, implying that $Fe_2O_3$, carbon, and $SnO_2$ were uniformly combined (Fig. 2h, i). The microstructures of the melamine sponge, and $Fe_2O_3$/C@$SnO_2$ (3:1:4)/ melamine sponge were characterized by SEM. The melamine sponge was found to have a porous and cellular-like structure with interconnected tetrapod-shaped frameworks. The frameworks width was about 4.3 μm (Supplementary Fig. 3a, b). The $Fe_2O_3$/C@$SnO_2$/ melamine sponge sample maintained the porous and interconnected structure. Due to the coated $Fe_2O_3$/C@$SnO_2$ layer, the surface of the sponge was found to be slightly rough, the framework width was about 4.30–7.34 μm while the thickness of sensing layer was about 0–3.04 μm (Supplementary Fig. 3d).

Furthermore, to evaluate the composition of $Fe_2O_3$/C and $Fe_2O_3$/C@$SnO_2$ nanostructures, the XPS technique was used. The full spectrum characteristic peaks were composed of Fe 2p, C 1s, O 1s, and Sn 3d states, as displayed in Fig. 3a. The peaks at 712.2 and 725.6 eV were ascribed to the Fe $2p_{3/2}$ and Fe $2p_{1/2}$, while the peak at 715.9 eV was attributed to Sn $3p_{3/2}$[31]. The $Fe_2O_3$/C@$SnO_2$ sample shows the Sn $3d_{3/2}$ and Sn $3d_{5/2}$ at around 493.3 and 484.9 eV, with a spin-orbit splitting of 8.4 eV, in concordance with the previously reported energy values for $SnO_2$[32].

**Sensing properties of the $Fe_2O_3$/C@$SnO_2$ pressure sensor.** To measure the piezoresistive characteristics of the pressure sensors, we setup a custom-made system composed of a universal testing machine and a digital source meter. Sensitivity was calculated using the formula: $S = (\Delta I/I_{unloading})/\Delta p$, where $\Delta I = I_{loading} - I_{unloading}$ represents the relative current change, $\Delta p$ refers to the change of pressure. Figure 4a, Supplementary Fig. 4, and Supplementary Table. 1 show the obtained measurements. The sensitivity of $Fe_2O_3$/C@$SnO_2$ pressure sensor was higher than that of $Fe_2O_3$, $Fe_2O_3$/C, and $SnO_2$@C pressure sensors. The sensitivity of $Fe_2O_3$/C@$SnO_2$ (3:1:4) sensor was $S_1 \sim 680$ kPa$^{-1}$ when the pressure was below 10 kPa, $S_2 \sim 98$ kPa$^{-1}$ when the pressure was ranged from 10 to 50 kPa, and $S_3 \sim 35$ kPa$^{-1}$ when the pressure was ranged from 50 to 150 kPa. This sensitivity is higher than that of ZnO sea-urchin-like, carbon sea-urchin-like, Ag/Au sea-urchin-like, and other pressure sensors (Supplementary Table 2). Even through the sea-urchin-like $Fe_2O_3$ structure promoted signal transduction and protected the $Fe_2O_3$ needles from mechanical breakage, the sensitivity of the sensor (3 kPa$^{-1}$) was still low, because $Fe_2O_3$ has poor conductivity. In addition, current changes were still small even under the high pressure. Supplementary Fig. 4a and Supplementary Table. 1 shows the current response of the pressure sensors under different mass

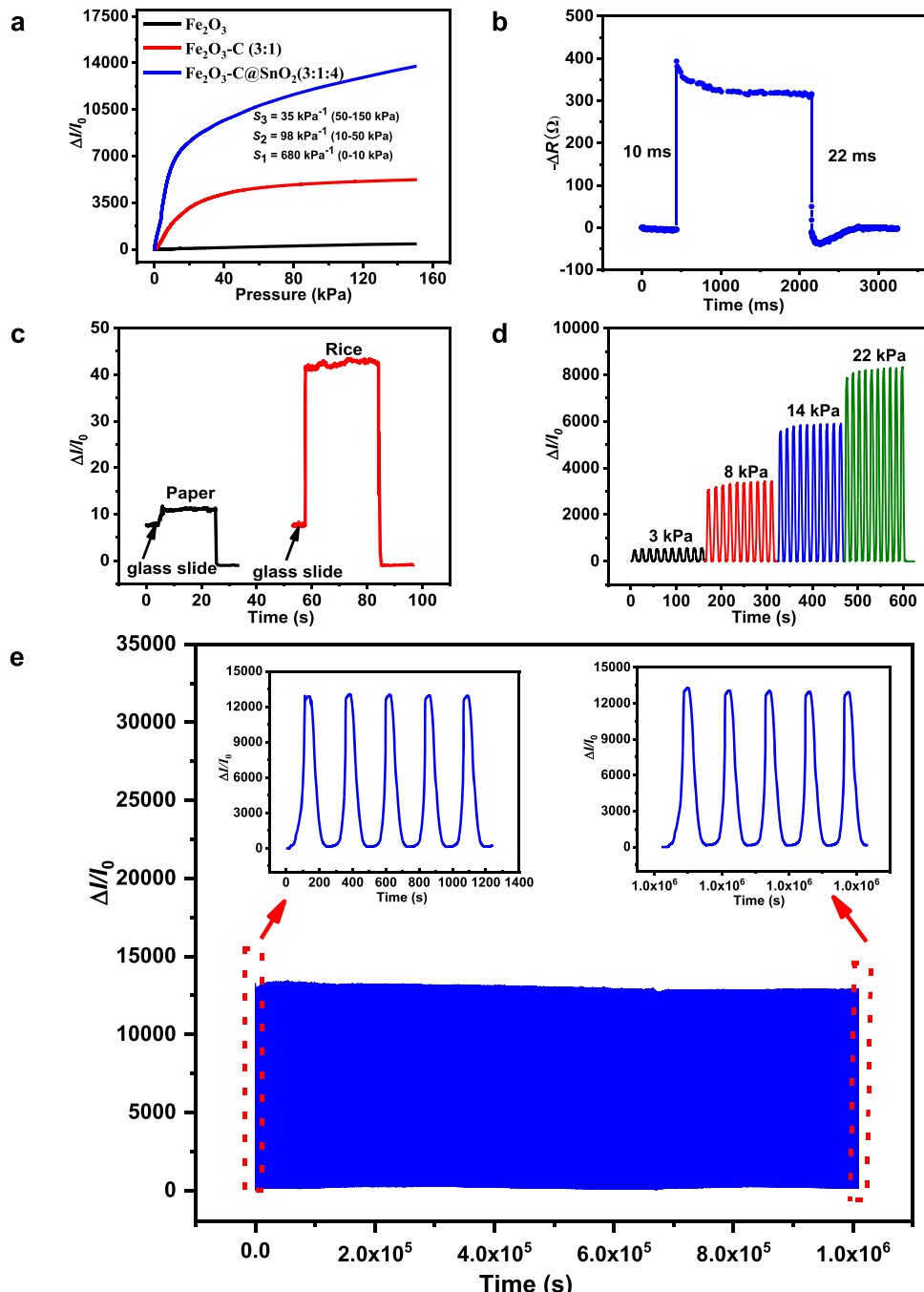

**Fig. 4 Pressure-sensing characterizations. a** The sensitivity of $Fe_2O_3$, $Fe_2O_3$/C with the mass of ratio of (3:1), and $Fe_2O_3$/C@$SnO_2$ with the mass of ratio of 3:1:4 based sensors. **b** Response time of $Fe_2O_3$/C@$SnO_2$ (3:1:4) pressure sensor. **c** Detection of low pressure: current curve of the proposed $Fe_2O_3$/C@$SnO_2$ (3:1:4) pressure sensor pressed by paper and rice grain. **d** The current response due to increased pressures under loading and unloading. **e** Stability performance of the $Fe_2O_3$/C@$SnO_2$ (3:1:4) pressure sensor with loading-unloading of more than 3500 cycles.

ratios of $Fe_2O_3$ and carbon. The mass ratio of 3:1 exhibited the highest sensitivity (203 kPa$^{-1}$) compared to the other lower ratios (2:1, 1:1 and 1:2) and to the higher one (4:1), let alone the pure carbon. The high sensitivity of the $Fe_2O_3$/C pressure sensor can be attributed to several factors. The microfibers of sponge are composed of nanocomposites, therefore, as the contact area is increased, there is corresponding an increase of current (Supplementary Fig. 12). Moreover, when carbon is added into the $Fe_2O_3$ system, the $Fe_2O_3$/C nanocomposite exhibits a larger current variation compared to that of pure carbon. Under the constant mass of $Fe_2O_3$, and the increased amounts of carbon, the sensitivity of the

$Fe_2O_3$/C pressure sensor increases due to an increase of the conductive path. However, excessive addition of carbon to the sensor may significantly increase conductivity (when the mass ratio of carbon and $Fe_2O_3$ exceeds 1:3), thus making it a good conductor, and in return affecting its increase in the corresponding conductive pathways. Here, we noted that the addition of carbon greatly improved the sensitivity of the sensor in the pressure range within 50 kPa, but did not improve the sensitivity in the pressure range over 50 kPa (Supplementary Fig. 4a).

To improve the sensitivity of the pressure sensor under a high-pressure range (over 50 kPa), $Fe_2O_3$/C (3:1) was further

combined with $SnO_2$. The addition of $SnO_2$ nanoparticles improve the sensitivity of the sensor in the low-pressure range (within 50 kPa) and improves its sensitivity in the pressure range (over 50 kPa) (Supplementary Fig. 4b and Supplementary Table. 1). Furthermore, when the two semiconductors were brought in contact and subjected to high-temperature calcination, the band alignment occurs driven by the equilibration of the Fermi level (Supplementary Fig. 10)[33,34]. Consequently, an n-n type heterostructure was formed between $Fe_2O_3$ and $SnO_2$, which promoted the transfer of electrons from $Fe_2O_3$ to $SnO_2$, thus enhancing the conductivity of the pressure sensor. Supplementary Fig. 2i shows that the $SnO_2$ nanoparticles grew on the (104) surface of the $Fe_2O_3$, along the direction of (110) $SnO_2$, thereby a forming (104) $_{Fe_2O_3}$/ (110) $_{SnO_2}$ heterojunction.

To further prove that the formation of heterojunction in $Fe_2O_3$/$C@SnO_2$ nanocomposites improved the sensing performance of the $Fe_2O_3$/$C@SnO_2$ pressure sensor. The sensing performances of the $Fe_2O_3$/$SnO_2$(3:4) and $Fe_2O_3$/$C@Fe_2O_3$(3:1:4) pressure sensors were measured (Supplementary Fig. 5a–c). The sensitivity of $Fe_2O_3$/$SnO_2$ (3:4) sensor was $S_1 \sim 8.5$ kPa$^{-1}$ when the pressure was below 10 kPa, $S_2 \sim 8.5$ kPa$^{-1}$ when the pressure was ranged from 10 to 50 kPa, and $S_3 \sim 8$ kPa$^{-1}$ when the pressure ranged from 50 to 150 kPa. It was higher than those of single $Fe_2O_3$ ($S_1 \sim 3$ kPa$^{-1}$, $S_2 \sim 3$ kPa$^{-1}$, and $S_3 \sim 2$ kPa$^{-1}$) and single $SnO_2$ pressure sensors ($S_1 \sim 1$ kPa$^{-1}$, $S_2 \sim 1$ kPa$^{-1}$, and $S_3 \sim 0.6$ kPa$^{-1}$). In addition, the sensitivity of $Fe_2O_3$/$C@Fe_2O_3$ (3:1:4) pressure sensor was $S_1 \sim 70$ kPa$^{-1}$ when the pressure was below 10 kPa, $S_2 \sim 9$ kPa$^{-1}$ when the pressure ranged from 10 to 50 kPa, and $S_3 \sim 2$ kPa$^{-1}$ when the pressure ranged from 50 to 150 kPa (Supplementary Fig. 5c). The sensitivity of $Fe_2O_3$/$C@Fe_2O_3$ (3:1:4) pressure sensor was lower than that of $Fe_2O_3$/$C@SnO_2$ (3:1:4) pressure sensor ($S_1 \sim 680$ kPa$^{-1}$, $S_2 \sim 98$ kPa$^{-1}$, and $S_3 \sim 35$ kPa$^{-1}$) (Supplementary Fig. 5b). The images of $Fe_2O_3$/$C@Fe_2O_3$ (3:1:4) reflect a typical sea-urchin-like structure (Supplementary Fig. 6). These findings show that the $Fe_2O_3$/$C@SnO_2$ heterostructure can improve the sensing performance of pressure sensor.

Collectively, the synergistic effects of the three structures ($Fe_2O_3$/C, $Fe_2O_3$/$SnO_2$ and $SnO_2@C$) improved the limited pressure response range of a single structure. Notably, the content of $SnO_2$ in $Fe_2O_3$/$C@SnO_2$ was associated with significantly improved the performance of the pressure sensor as shown in Supplementary Fig. 4b. In addition, when more $SnO_2$ ($Fe_2O_3$/$C@SnO_2$(3:1:8)) was added in the synthesis, the polymerization of $SnO_2$ nanoparticles occurred due to their high surface energy, and subsequently leading to a non-uniformed distribution of $SnO_2$ nanoparticles in $Fe_2O_3$/$C@SnO_2$. However, when less $SnO_2$ ($Fe_2O_3$/$C@SnO_2$(3:1:1)) was added in the synthesis, less accumulation layer was formed on $SnO_2$, thereby affecting the conductivity of the pressure sensor. Therefore, the obtained $Fe_2O_3$/$C@SnO_2$(3:1:8) and $Fe_2O_3$/$C@SnO_2$(3:1:1) had a lower sensitivity compared to $Fe_2O_3$/$C@SnO_2$(3:1:4). Moreover, $Fe_2O_3$/$C@Sb_2O_3$ (3:1:4) was synthesized and characterized to verify whether the ternary structure has a certain universality in improving the sensitivity and expanding the pressure working range of the piezoresistive pressure sensor. The sensitivity of $Fe_2O_3$/$C@Sb_2O_3$ (3:1:4) pressure sensor was found to be $S_1 \sim 303$ kPa$^{-1}$ when the pressure was below 10 kPa, $S_2 \sim 180$ kPa$^{-1}$ within the pressure range of 10–50 kPa, and $S_3 \sim 13$ kPa$^{-1}$ within the pressure range of 50–150 kPa (Supplementary Fig. 11). The experimental results show that the ternary structure has a certain universality in improving the sensitivity and in expanding the pressure working range of the piezoresistive pressure sensor.

To determine whether the sensing performance of the pressure sensor is affected by humidity, we tested the sensing performance of the $Fe_2O_3$/$C@SnO_2$ (3:1:4) pressure sensor at room temperature with a relative humidity (RH) of 73%, 85%, and 95%.

Supplementary Fig. 7 shows that the sensing performance of the $Fe_2O_3$/$C@SnO_2$ (3:1:4) pressure sensors did not change with increasing the relative humidity, implying that their sensing performance was independent of relativity humidity. In addition, variations of current ratios with the pressure of the sponges with different areas and thicknesses was similar (Supplementary Fig. 8a–f), which indicates that the sensing performance of the $Fe_2O_3$/$C@SnO_2$ sponge is independent of its area and thickness. All results show that the $Fe_2O_3$/$C@SnO_2$ pressure sensors are stable.

Besides, we further assessed the low limit detection of the $Fe_2O_3$/$C@SnO_2$ (3:1:4) pressure sensor as outlined in Fig. 4c. To evaluate the low limit detection of the pressure sensor, a weight of 4.2 g glass slide was placed on the pressure sensor. Thereafter, a paper (~0.52 pa, $m = 0.0107$ g, $S = 1 \times 2$ cm$^2$) and rice ($m = 0.0116$ g) were put on the glass slide. The glass slide served two purposes; completing the contact between the electrode and the pressure sensor, and stabilizing the current more. The response time of the $Fe_2O_3$/$C@SnO_2$ (3:1:4) pressure sensor is shown in Fig. 4b. When compressing the sponge pressure sensor by 0.02 mm at a speed of 500 mm per min, the response time and recovery time of the $Fe_2O_3$/$C@SnO_2$ (3:1:4) pressure sensor was 10 and 22 ms, respectively. The hysteresis of the recovery time of this pressure sensor may be attributed to the sponge substrate. To evaluate the stability of the pressure sensor under different pressures, the $Fe_2O_3$/$C@SnO_2$ (3:1:4) pressure sensor was set under various pressure values of 3, 8, 14, and 22 kPa (Fig. 4d). These findings reveal that the current gradually increases with increasing pressure. Therefore, the $Fe_2O_3$/$C@SnO_2$ (3:1:4) pressure sensor can clearly distinguish different pressure values. Furthermore, repeated compression/release test over 3500 cycles with a peak pressure of 110 kPa was performed (Fig. 4e). The insets revealed the 5 cycles of the current response at the inception (left) and termination (right) of the stability test, while, the device indicated a stable signal without offset during the cycles test, thereby reflecting that the performance of the sensor was stable under long cycles and high pressure. The SEM images of sponge in the original state and compression/release over 3500 cycles state are shown in Supplementary Fig. 2g, h. Compared to the original state, the micromorphology of $Fe_2O_3$ did not exhibit any change when the sensor was repeatedly compressed/released under high pressure. However, only a small part of the $Fe_2O_3$ needles was detached from the sea-urchin-shaped microspheres. In this work, the good stability of the sensor could be attributed to two reasons; the tapering geometry of $Fe_2O_3$ protects the bristle from mechanical breaking[21,22], and there is a lot of space between the $Fe_2O_3$ needle, which can allow carbon and $SnO_2$ nanoparticles to easily absorb onto the interval gap, thereby protecting the integrity of the $Fe_2O_3$ needle structure.

**Extremely high-pressure resolution.** A key feature of the $Fe_2O_3$/$C@SnO_2$ (3:1:4) pressure sensor is high sensitivity in wide pressure ranges. Therefore, we evaluated the sensitivity of this pressure sensor under high pressure by subjecting it to different pressure values at 1.5, 10 and 50 kPa (Fig. 5a–c). First, the pressure sensor was subjected to the set pressure value, followed by consecutive addition of three coins, each weighing about 3.19 g, equivalent to a pressure of 86 Pa. Each pressure increment caused a stepwise increment in current, with the current signal being stable. In another experiment, a pressure sensor with a volume of $V = 19 \times 19 \times 4$ mm$^3$ was placed under the front wheel of a car (the weight of the car was 1670 kg) as shown in Fig. 5d. Thereafter, a carton of milk weighing 4 kg put on the driving seat of the car and then taken away as indicated in Fig. 5e. The changes in current were successfully detected. When the male

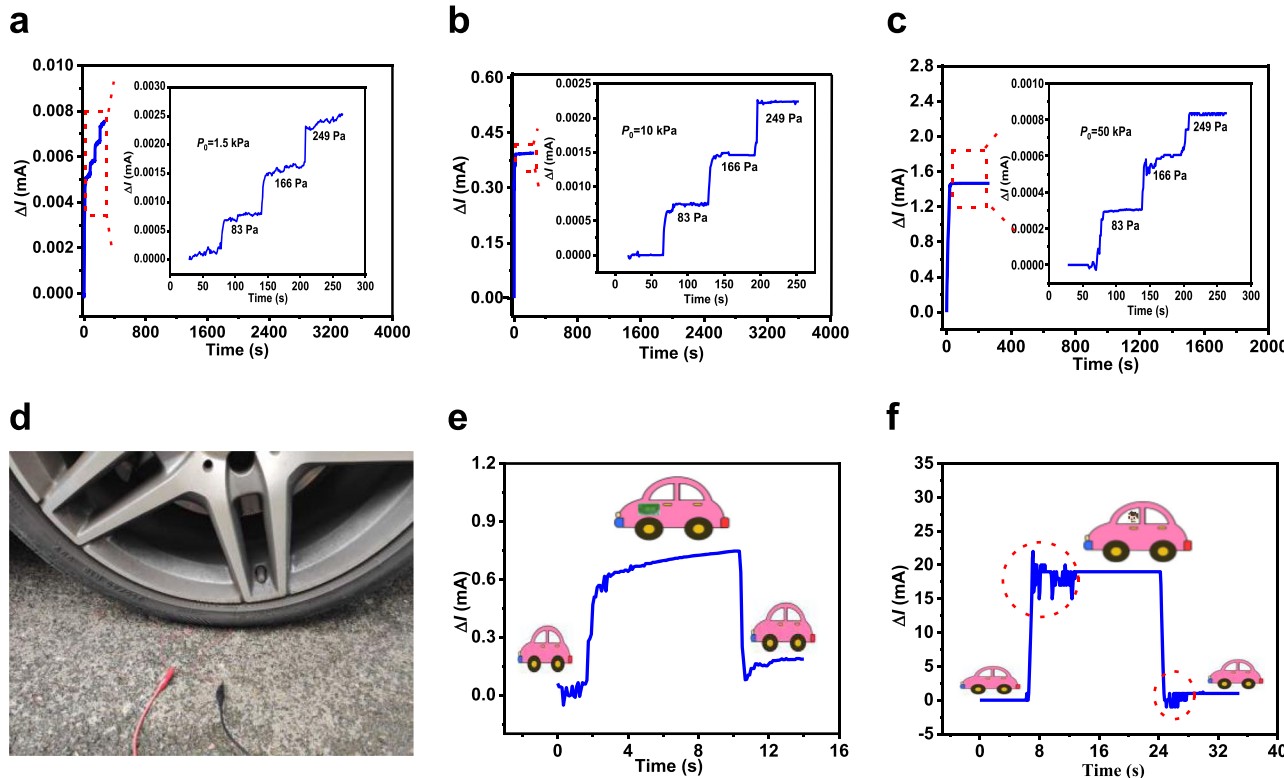

**Fig. 5 Extremely high-sensing resolution of the Fe₂O₃/C@SnO₂ pressure sensor.** Detection of micro pressure under loading pressures of **a** 1.5 kPa, **b** 10 kPa, and **c** 50 kPa. **d** Experimental setup of a car with a Fe₂O₃/C@SnO₂ (3:1:4) pressure sensor attached under a front tire. **e** Current signals corresponding to an unloaded, loaded, and unloaded 4 kg carton of milk on the driving seat of the car. **f** Current signals corresponding to a 73 kg male passenger getting into and out of the car.

passenger with a weight of 73 kg gets into or out of the car, the current changed significantly (Fig. 5f). The circled vibrations in Fig. 5f reveal that the pressure sensor can accurately capture the movement of the male passenger getting on or off the car. The sensitivity of the pressure sensor under high pressure was also tested by the tensile test equipment. The sensing performance of device under a pressure of $P_0 = 210$ kPa is shown in Supplementary Fig. 9a, b. During the test, the device was first subjected to the reference pressure. Then, the pressure sensor was added with pressure of 2.8 and 25 kPa, respectively. It was found that the pressure sensor still had high sensitivity under the high pressure.

**Wearable device demonstration**. Due to the high sensitivity, fast response time, and broad pressure regime of the pressure sensor (Fe₂O₃/C@SnO₂ (3:1:4)), it can be applied in various fields. For instance, it can be used to detect voice, wrist pulse, and human motion activities. The Fe₂O₃/C@SnO₂ (3:1:4) pressure sensor was attached to the skin with the help of polyimide (PI) tape for all human body interactions. Figure 6a shows the real-time wrist pulse as detected using the Fe₂O₃/C@SnO₂ (3:1:4) pressure sensor. The testing curves revealed strong characteristic peaks of the human sphygmic waveforms, with the pulse rate being about 73 times per min, which is the normal level. Based on the excellent performance of the Fe₂O₃/C@SnO₂ (3:1:4) pressure sensor, it can be used for monitoring human health. This pressure sensor was also attached to the human throat to monitor and distinguish subtle differences of muscle motions near the throat, when the words one, two and three were spoken (Fig. 6b). Interestingly, this technique can be used by deaf and mute people who are unable to speak. It is well-known that their vocal cords can vibrate, and thus, the vibration produced can be transformed into the required sound[7]. Moreover, the pressure sensor was mounted on the cheek

to monitor the occlusion movements of humans, as displayed in Fig. 6c. Upon occlusion, the current was found to have significantly changed, proving the excellent performance of the sensor. Moreover, the pressure sensor was also attached to the arm to detect radial muscle contractions, which occur when making a fist (Fig. 6d). When the tester made a fist, the current signal and compression of the pressure sensor increased, illustrating its potential application in physical training and curing muscle damage. The response of the pressure sensor for continuously bending six different motions of the finger is presented in Fig. 6e. It showed different responsive current signals for different motions of the finger. In particular, the current signal exhibited a slight increase when the finger bent in small-scale (motion-I, motion-II, and motion-III), whereas larger-scale bending led to a sharp increase of current value (motion-IV, motion-V, and motion-VI). Large-scale movements resulted in a strong compression of the pressure sensor, thereby forming more conductive pathways. These findings imply that the pressure sensor can precisely distinguish different-scale motions of the finger. Figure 6f shows that the pressure sensor was also mounted on foot using tape to monitor walking states. The response signals of walking motion were stable and repetitive, suggesting that it can be applied in gait recognition and motion monitoring. These outcomes show that the Fe₂O₃/C@SnO₂ pressure sensor has broad application prospects in medical health and wearable electronic devices.

## Discussion
In summary, we present a sea-urchin-shaped microstructure Fe₂O₃/C@SnO₂, which was synthesized using a simple and environmentally friendly hydrothermal method. Moreover, we provide a pressure sensor with high sensitivity and a large

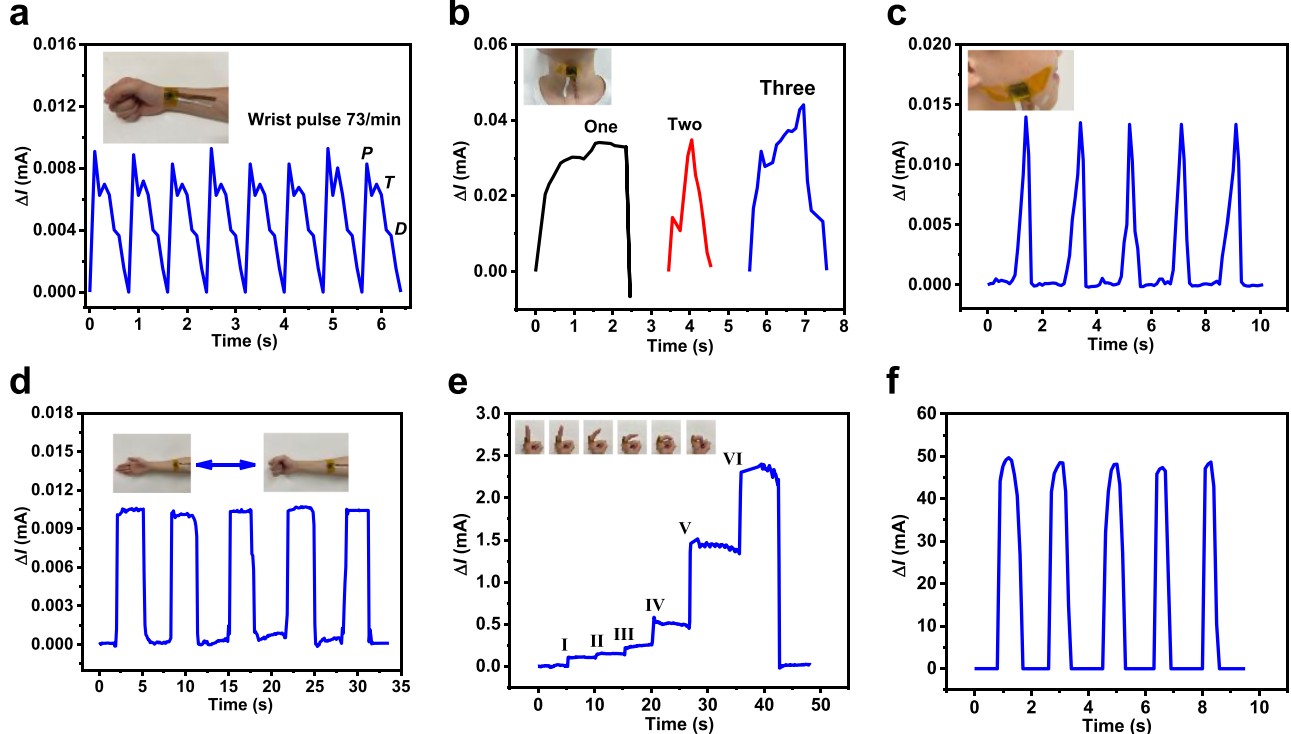

**Fig. 6 Wearable demonstration. a** The current response caused by the arterial pulse waves with the sensor attached to the wrist. **b** The recorded current signal versus time pronouncing. Finally, **c** the signal variations of relative current corresponding to different occlusion, **d** human palm, **e** finger motion, and **f** walking.

working range based on a simple dip-coating process method. The $Fe_2O_3/C@SnO_2$ pressure sensor exhibits high sensitivity ($680 \, kPa^{-1}$), fast response (10 ms), broad range (up to 150 kPa), and good reproducibility (over 3500 cycles under a pressure of 110 kPa). Interestingly, multiple human physiological activities (such as pulse, pronunciations, joint bending, and walking, among others) could be monitored by the $Fe_2O_3/C@SnO_2$ pressure sensor. Based on the above excellent performances of this device, it has significant implications in wearable electronics, health monitoring, and measuring pressure distribution.

## Methods

**Synthesis of $Fe_2O_3$**. The method used for the synthesis of $Fe_2O_3$ used in this study[35]. In particular, the $Fe_2O_3$ was synthesized through a hydrothermal process, where 0.405 g FeCl₃.6H₂O (Aladdin) and 0.205 g Na₂SO₄ (Aladdin) were first dissolved in distilled water (30 mL) and stirred for 10 min. Then, the mixture was heated at 120 °C for 6 h in a Teflon-lined stainless-steel autoclave. After cooling, filtering, drying, and thermal annealing at 400 °C for 3 h under air, $Fe_2O_3$ was obtained.

**Preparation of $Fe_2O_3/C$ composites**. Besides adding different masses of carbon, the same process was used for the synthesis of $Fe_2O_3/C$ composites.

**Synthesis of $Fe_2O_3/C@SnO_2$ composites**. First, the $Fe_2O_3/C$ composite (0.145 g), Na₂SnO₃ (0.145 g, 0.036 g, 0.290 g) (Aladdin), and urea (1.16 g, 0.290 g, 2.320 g) (Aladdin) were dissolved in ethanol/H₂O solution and kept $V_{ethanol}$ : $V_{H2O}$ = 2:3. After stirring for 15 min, the mixture was heated at 180 °C for 6 h in a Teflon-lined stainless-steel autoclave. After natural cooling, the $Fe_2O_3/C@SnO_2$ composites were washed several times using distilled water and absolute ethanol. Finally, the $Fe_2O_3/C@SnO_2$ composites were dried at 60 °C for 12 h.

**Synthesis of $Fe_2O_3/C@Sb_2O_3$ composites**. The $Fe_2O_3/C@Sb_2O_3$ composites was synthesized through a hydrothermal process, where 0.34 g $Fe_2O_3/C$ and 0.34 g SbCl₃ (Aladdin) were first dissolved in ethanol (40 mL) and stirred for 30 min. Afterward, the mixture was heated at 140 °C for 12 h in a Teflon-lined stainless-steel autoclave. Lastly, after cooling, filtering, drying, and thermal annealing at 400 °C for 3 h under air, $Fe_2O_3/C@Sb_2O_3$ was obtained.

**Preparation of conductive sponges and electrode**. A melamine sponge was cut into a cuboid with a length of 19 mm, width (19 mm), and height (4 mm). Then, it was washed several times using ethanol and dried at 45 °C. Thereafter, $Fe_2O_3/C@SnO_2$ ($SnO_2@C$, $Fe_2O_3/C$, $Fe_2O_3$, $SnO_2$, and C) and Polyvinylidene fluoride (PVDF) binder (Aladdin) were dissolved in N-methyl pyrrolidone (NMP) (Aladdin) in a weight ratio of 10:1 and mixed to form a slurry. Next, the melamine sponge strip was immersed in the slurry until it was full and dried at 45 °C in a vacuum. A copper wire was then fixed on the polyimide (PI) film (Kapton) substrate coated with laser-induced graphene (LIG) with the silver paste, where the PI-LIG layer acted as the electrode of the device. The copper tape was fixed on the PI film to flat the PI film.

**Structural characterization and the performance of sensors**. The crystal structures of the samples were explored using X-ray diffraction (XRD, PANalytical X'Pert Powder), while the morphology was characterized using scanning electron microscopy (SEM; Quattro S), high-angle annular dark-field scanning transmission electron microscopy (HAADF-STEM), elemental mapping and transmission electron microscopy (TEM; Talos F200S). Binding energy of the products was investigated using X-ray photoelectron spectroscopy (XPS; ESCALAB250Xi). Moreover, the loading of pressure was examined using a universal testing machine (ETM-5038, Shenzhen Wance Testing Machine Co., Ltd.), while the electrical signals of the pressure sensors were recorded at the same time using a digital meter (Keithley 2450, Optoelectronic Technology & Systems) at a constant voltage of 0.1 V. Finally, to assess the response time of the pressure sensor, a multimeter (Keithley DAQ6510, Optoelectronic Technology & Systems) was used.

## Data availability

The data used this study are available from the corresponding author upon reasonable request.

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

## Acknowledgements

This research was supported in part by the National Natural Science Foundation of China under Grant 62071073, in part by the National Key Research and Development Program under Grant 2018YFB2100100, in part by the Fundamental Research Funds for Central Universities under Grant 2019CDJGFGD007, in part by the Key Science and Technology Program of Chongqing under Grant CSTC2017SHMS-ZDYFX0028, in part by the Technology Innovation and Application Project of Chongqing under Grant cstc2018jszx-cyzdX0111, in part by the China Postdoctoral Science Foundation under Grant 2019M663433 and in part by the Guangxi Key Laboratory of Manufacturing Systems and Advanced Manufacturing Technology under Grant 19-050-44-002 K.

## Author contributions

X.W. prepared the sample and wrote the manuscript. L.T., M.Y., Z.W., J.Y., X.C., and C.W. helped to modify the manuscript and design the experiments. D.X. and F.L. helped test.

## Competing interests

The authors declare no competing interests.

## Additional information



