## [Peer Review File · Nature Communications]

REVIEWER COMMENTS

Reviewer #1 (Remarks to the Author):

Wang et al. fabricated Fe₂O₃/C@SnO₂ urchin-like microstructures for pressure sensor application. Melamine sponge has been used for the backbone of pressure sensor body. Various compositions have been tested systematically to find the optimum composition. The application of the pressure sensor to monitor human motion and voice is quite interesting. I recommend the publication of the manuscript with major revision.

1. The optimized sensor shows three working ranges with different sensitivities. To ensure the super working range of the sensor, the sensitivity and the corresponding working range should be compared with previous reports.
2. SEM images of melamine sponge before and after dipping in the slurry should be presented. How thick is the sensing layer coated on the melamine sponge?
3. Photographs of the fabricated sensor should be presented. What is the thickness of the copper tape, the electrode?
4. How do you attached the sensor to the skin?
5. The sensor performance should be tested in humidity condition to know whether humidity affects the sensing performance.

Reviewer #2 (Remarks to the Author):

1. The authors have used a novel composite material for pressure sensing. The reported work displayed excellent sensitivity.
2. In Fig 4a, S1 S2 S3 ranges are not clearly represented.
3. The authors claim that the working mechanism of pressure sensor is due to change in contact area of the nanocomposite coated sponge structure with respect to applied pressure. It is not clear how formation of Fe₂O₃/C@SnO₂ heterostructure helps in increasing the pressure range? Does growth of SnO₂ on Fe₂O₃ nanoneedles improve its mechanical properties?
4. Why Fe₂O₃/C response is constant at higher pressure range?
5. What are the dimensions of melamine sponge substrate used in the experiment? Does the dimension affect the pressure sensing range?

Reviewer #3 (Remarks to the Author):

The author has carried out good works. The sensor was properly synthesized and characterized with various type of load. The author got good sensitivity at low pressure range. But I have few concern about the paper and comments are as follows.

1. The paper title is not appropriate because of the word "super working range".
2. Under the headline of structural characterisation, the first para is not appropriate.
3. In the comparative study of graph 2 a, the first peak of composite (Fe₂O₃/C@SnO₂ 3:1:4) seems to be broadened instead of sharp. [In Page 6]
4. In page 7, 3rd line: "Either between or adjacent". Both words should not be written in a statement.

5. In page 3: it's mentioned that "carbon encloses the particles" and in page 7: it is mentioned that "carbon material was embedded in the gap". Embedding and enclosing have different meaning.
6. In page 8
 - I. In Figure 3 there is no graph of carbon present.
 - II. Graph 3 c and 3 d are same.
 - III. Inside the graph "ev" written wrong. "v" should be capital.
 - IV. In second last line 4844.9 eV has been written which should be 484.9
7. In Fig 4 c, Caption: it should be detection of low pressure not weak pressure.
8. Page 10 : 1st line: Of pressure sensor. "is" should be omitted.
9. Page 11 : 1st para last line: The statement should not include obviously.
10. Page 12 : 1st para, Last 5th line from bottom: Either it should be "sensor is" or "sensors are".
11. Page 13: Recovery time seems to be more than 22 ms. Why author have not considered the dip part (after 2000 ms) in the Fig. 4 d, before graph returns to its original state?
12. In section "High Pressure resolution" instead of using front wheel of a car author should use tensile testing equipment to show the sensor is having resolution in high pressure load also.

For the paper to be the part of Nature communication publication, it should have innovative approach or new phenomena to be observed in the materials. In this paper author achieve good sensitivity of the sensor in wide pressure range. This paper lack to show the innovative approach or novelty. Therefore this paper should be rejected for publication in this journal.

List of revisions made

1. According to referee 1's and referee 3's comments, this paper lack to show the innovative approach or novelty.

The sensitivity and the corresponding working range have been compared with previous reports.

We have added this Table in the supporting information of the revised paper (Supplementary Table 2, which is in the page 4).

Supplementary Table 2 Sensitivity summary of pressure sensor in reference

Materials	Nanostructure	Sensitivity (kPa ⁻¹)	Reference
ZnO/PDMS	Sea urchin-like	75 - 121 (0 - 200 Pa)	23 (Nat. Commun. 2018)
Au/Ag/PU	Sea urchin-like	2.46 (0 - 1 kPa) 0.52 (1 -8.2 kPa)	24 (Adv. Mater. 2016)
C/PDMS	Sea urchin-like	263 at 1 Pa	25 (Nat. Commun. 2020)
Mxene/Reduced Graphene Oxide Aerogel	Naosheets	0.55 (23 - 982 Pa) 3.81 (982Pa - 10 kPa) 2.52 (10 - 30 kPa)	36 (ACS Nano 2018)
Carbon Black@Polyurethane Sponge	Nanosheets	0.068 (0-2 kPa) 0.023 (2-10 kPa) 0.036 (10-16 kPa)	37 (Adv. Funct. Mater. 2018)
MXene/Sponge	Nanosheets	147 (0-5 kPa) 442 (5-20 kPa)	38 (Nano Energy .2018)
Graphene/Eco-flex	Triode-mimicking	4.68 (0-150 kPa) 11.09 (150-200 kPa)	39 (ACS Nano 2020)
CNT/cotton textile	Nanowires	14.4 (0-3.5 kPa) 7 (3.5-15 kPa)	40 (Adv. Mater. 2017)
MXene/PLA	Nanosheets	0.55 (0-3 kPa) 3.81 (3-10 kPa)	41 (Nano lett. 2019)
Fe ₂ O ₃ /C@SnO ₂ /Sponge	Sea urchin-like	680 (0-10 kPa) 98 (10-50 kPa) 35 (5-150 kPa)	In this work

This article is innovation as follows:

(1) Compared with previous work (as shown in table), the Fe₂O₃/C@SnO₂ pressure sensor exhibited high sensitivity (680 kPa⁻¹), broad range (up to 150 kPa) and good reproducibility (over 3500 cycles under a pressure of 110 kPa).

(2) We have found that the performance of pressure sensors can be improved when the heteostructure is used. And the heteostructure has rarely been employed in fabricating flexible pressure sensors. A depletion region and bands bending occurs in the contact sections of the heterojunction, which induces lower interfacial resistance and promoting of the charge

transport/transfer. The heterostructure is formed between Fe_2O_3 and SnO_2 , which promotes the transfer of electrons from Fe_2O_3 to SnO_2 , thus enhancing the sensing performance of the pressure sensor.

(3) The sea urchin-like ternary nanocomposite $\text{Fe}_2\text{O}_3/\text{C}@\text{SnO}_2$ also contains $\text{Fe}_2\text{O}_3/\text{C}$ and $\text{SnO}_2@\text{C}$ hybrid structures. A ternary composite structure is rarely used in pressure sensors. The sea urchin-like Fe_2O_3 structure promotes signal transduction and protected Fe_2O_3 needles from mechanical breaking. Compared with single Metal oxide semiconductor, metal oxide semiconductor/conductor interface piezoresistive effect is essentially favorable for the change of contact area, which thus leads to big output current change. So, $\text{Fe}_2\text{O}_3/\text{C}$ and $\text{SnO}_2@\text{C}$ hybrid structures can improve the sensing performance of pressure sensor.

We also revised the second paragraph of the introduction section

The initial version was: There are two widely researched metal oxide including Fe_2O_3 and SnO_2 , because of their low-cost, environmental friendliness, and natural abundance. Studies have reported that coupling metal oxide and carbon compounds to form metal oxide/C nanocomposite, may improve photocatalytic ²⁶ and electrochemical performances ²⁷. Despite metal oxide/C nanocomposite processing large specific surface area and strong conductivity, they have rarely been employed in fabricating flexible pressure sensors.

In this work, we proposed a nanostructure design of materials with ultra-sensitivity for an ultra-broad-range pressure sensor. Particularly, this strategy involves the use of acetylene black carbon as a carrier due to its strong conductivity and high specific surface. The acetylene black carbon encloses particles Fe_2O_3 , thereby forming $\text{Fe}_2\text{O}_3/\text{C}$ structure. Furthermore, one part of SnO_2 nanoparticles were dispersed into the carbon layer and formed $\text{SnO}_2@\text{C}$ structures, whereas its other part of nanoparticle adhered to the surface of Fe_2O_3 needles and formed $\text{Fe}_2\text{O}_3/\text{SnO}_2$ heterostructures. Carbon improves the conductivity of a single metal oxide. Collectively, the synergy of the three structures ($\text{Fe}_2\text{O}_3/\text{C}$, $\text{Fe}_2\text{O}_3/\text{SnO}_2$ and $\text{SnO}_2@\text{C}$) improved the limited pressure response range of a single structure. Notably, the $\text{Fe}_2\text{O}_3/\text{C}@\text{SnO}_2$ (3:1:4) pressure sensor exhibited high sensitivity (680 kPa^{-1}), fast response (10 ms), broad range (up to 150 kPa) and good reproducibility (over 3500 cycles under a pressure of 110 kPa).

The revision version was: High sensitivity can be obtained under two conditions: low initial

current and large output current change under certain pressure²⁶. The conductivity of semiconductor is considerably low, so the initial current could achieve low level. In addition, semiconductor/conductor interface piezoresistive effect is essentially favorable for the change of contact area, which thus leads to big output current change²⁶. A depletion region and bands bending occurs in the contact sections of the heterojunction, which induces lower interfacial resistance and promotes of the charge transport/transfer²⁷. Therefore, heterojunction have been used in many modern devices, including light emitting diodes (LEDs), photodetectors and solar cells²⁸⁻³⁰. Therefore, when the metal oxide semiconductor/C composite structure and the heterostructure are used in the pressure sensor, the sensing performance of the pressure sensor may be improved.

In this work, we proposed a pressure sensor with nanostructure design of materials and the nanostructure contains metal oxide semiconductor/C composite structure and the heterostructure. The pressure sensor exhibited ultra-sensitivity and ultra-broad-range, when fabricated using the new nanostructure. We choose Fe_2O_3 and SnO_2 as sensing materials, because of their low-cost, environmental friendliness, and natural abundance. Sea urchin-like Fe_2O_3 is synthesized via a hydrothermal method. Particularly, this strategy involves the use of acetylene black carbon as a carrier due to its strong conductivity and high specific surface. One part of acetylene black carbon encloses particles Fe_2O_3 , whereas its other part of carbon materials was embedded in the gap Fe_2O_3 needles, forming $\text{Fe}_2\text{O}_3/\text{C}$ structure. Furthermore, one part of SnO_2 nanoparticles were dispersed into the carbon layer and formed $\text{SnO}_2@\text{C}$ structures, whereas its other part of nanoparticle adhered to the surface of Fe_2O_3 needles and formed $\text{Fe}_2\text{O}_3/\text{SnO}_2$ heterostructures. Carbon improves the conductivity of a single metal oxide. Collectively, the synergy of the three structures ($\text{Fe}_2\text{O}_3/\text{C}$, $\text{Fe}_2\text{O}_3/\text{SnO}_2$ and $\text{SnO}_2@\text{C}$) improved the limited pressure response range of a single structure. Notably, the $\text{Fe}_2\text{O}_3/\text{C}@\text{SnO}_2$ (3:1:4) pressure sensor exhibited high sensitivity (680 kPa^{-1}), fast response (10 ms), broad range (up to 150 kPa) and good reproducibility (over 3500 cycles under a pressure of 110 kPa).

2. According to referee 1's and referee 2's comments, we have tested the sensor performance under different humidity conditions. We have tested the sensing performance of the $\text{Fe}_2\text{O}_3/\text{C}@\text{SnO}_2$ (3:1:4) pressure sensor with different dimensions. We also measured the pressure response curves of pressure sensors ($\text{Fe}_2\text{O}_3/\text{SnO}_2$ (3:4) and $\text{Fe}_2\text{O}_3/\text{C}@\text{Fe}_2\text{O}_3$ (3:1:4)) to further prove the improvement of the sensing performance of pressure sensor by heterojunction.

3. Supplementary Fig. 3 and supplementary Fig 5-9 has been added in page 3-5 in supplementary information. Supplementary Fig. 3-6 have been changed supplementary Fig. 4, supplementary Fig.10

Response to referees' comments

Response to referee 1's comments:

1. The optimized sensor shows three working ranges with different sensitivities. To ensure the super working range of the sensor, the sensitivity and the corresponding working range should be compared with previous reports.

Response:

Thanks very much for your valuable comment.

We have added this Table in the supporting information of the revised paper (Supplementary Table 2, which is in the page 6).

Supplementary Table 2 Sensitivity summary of pressure sensor in reference

Materials	Nanostructure	Sensitivity (kPa ⁻¹)	Reference
ZnO/PDMS	Sea urchin-like	75 - 121 (0 - 200 Pa)	23
Au/Ag/PU	Sea urchin-like	2.46 (0 - 1 kPa) 0.52 (1 - 8.2 kPa)	24
C/PDMS	Sea urchin-like	263 at 1 Pa	25
Mxene/Reduced Graphene Oxide Aerogel	Naosheets	0.55 (23 - 982 Pa) 3.81 (982Pa - 10 kPa) 2.52 (10 - 30 kPa)	36
Carbon Black@Polyurethane Sponge	Nanosheets	0.068 (0-2 kPa) 0.023 (2-10 kPa) 0.036 (10-16 kPa)	37
MXene/Sponge	Nanosheets	147 (0-5 kPa) 442 (5-20 kPa)	38
Graphene/Eco-flex	Triode-mimicking	4.68 (0-150 kPa) 11.09 (150-200 kPa)	39
CNT/cotton textile	Nanowires	14.4 (0-3.5 kPa) 7 (3.5-15 kPa)	40
MXene/tissue paper	Nanosheets	0.55 (0-3 kPa) 3.81 (3-10 kPa)	41
Fe ₂ O ₃ /C@SnO ₂ /Sponge	Sea urchin-like	680 (0-10 kPa) 98 (10-50 kPa) 35 (5-150 kPa)	In this work

2. SEM images of melamine sponge before and after dipping in the slurry should be presented.

How thick is the sensing layer coated on the melamine sponge?

Response:

Thanks very much for your valuable comment.

The SEM images have been presented in the supporting information of the revised paper (supplementary Fig. 3, page 3). The SEM images are as follows:

We have added this description “The microstructures of the Fe_2O_3 , $\text{Fe}_2\text{O}_3/\text{C}$ (3:1), and $\text{Fe}_2\text{O}_3/\text{C}@\text{SnO}_2$ (3:1:4) were characterized using SEM. The melamine sponge has a porous and cellular-like structure with the interconnected tetrapod-shaped frameworks; and the frameworks width is about 4.3 μm (Supplementary Fig. 3a-b). The $\text{Fe}_2\text{O}_3/\text{C}@\text{SnO}_2$ / melamine sponge sample maintains the porous and interconnected structure; the surface of the sponge becomes slightly rough with the coated $\text{Fe}_2\text{O}_3/\text{C}@\text{SnO}_2$ layer, and the framework width is about 4.30-7.34 μm . The thickness of sensing layer is about 0-3.04 μm (Supplementary Fig. 3d).” in the revised paper, line16, page 7.

3. Photographs of the fabricated sensor should be presented. What is the thickness of the copper tape, the electrode?

Response:

Thanks very much for your valuable comment.

The photograph of sensor is shown below:

The photograph has been added in the revised paper and named Fig. 1b. And we have added

this description “Fig. 1b shows the photograph of pressure sensor.” in the revised paper, line 18, page 4.

The thickness of the copper tape is 0.06 mm.

4. How do you attached the sensor to the skin?

Response:

Thanks for your comment.

As shown in the below figure, the $\text{Fe}_2\text{O}_3/\text{C}@/\text{SnO}_2$ pressure sensor is attached to the skin with polyimide (PI) tape. We have added this description “The $\text{Fe}_2\text{O}_3/\text{C}@/\text{SnO}_2$ (3:1:4) pressure sensor was attached to the skin with the help of polyimide (PI) tape for all human body interactions.” in the revised paper, line 1, page 18.

5. The sensor performance should be tested in humidity condition to know whether humidity affects the sensing performance.

Response:

Thanks very much for your valuable comment.

Sensitivities of $\text{Fe}_2\text{O}_3/\text{C}@/\text{SnO}_2$ (3:1:4) pressure sensor under the relative humidity (RH) of 73%, 85% and 95% at room temperature, as illustrated in supplementary Fig. 7, page 4.

Humidity (%)	Pressure (kPa)			
	Sensitivity (kPa ⁻¹)	0-10	10-50	50-150
Initial		680	98	35
73% RH		681	98	31
85% RH		684	94	31
95% RH		685	100	33

I'm sorry that we didn't record the relative humidity value in the environment in the manuscript. The relative humidity value of the sensitivity test in the manuscript is named "Initial". It can be seen in above figures that the performance of the Fe₂O₃/C@SnO₂ (3:1:4) pressure sensors is similar with increasing relative humidity levels. It indicates that the sensing performance of the Fe₂O₃/C@SnO₂ (3:1:4) pressure sensors is independent of relative humidity.

We have added this description "To analyze whether the sensing performance of pressure sensor is affected by humidity, we test the sensing performance of the Fe₂O₃/C@SnO₂ (3:1:4) pressure sensor at room temperature with a relative humidity (RH) of 73%, 85%, and 95%. It can be seen in supplementary Fig. 7 that the sensing performance of the Fe₂O₃/C@SnO₂ (3:1:4) pressure sensors is similar with increasing relative humidity levels, which indicates that the sensing performance of the Fe₂O₃/C@SnO₂ (3:1:4) pressure sensors is independent of relative humidity." in the revised paper, line 17, page 13.

Response to referee 2's comments:

1. The authors have used a novel composite material for pressure sensing. The reported work displayed excellent sensitivity.

Response:

Thanks to referee 2 for your positive comments.

2. In Fig 4a, S1 S2 S3 ranges are not clearly represented.

Response:

Thanks very much for your valuable comment.

Pressure ranges have been clearly represented in Fig. 4a, page 9.

3. The authors claim that the working mechanism of pressure sensor is due to change in contact area of the nanocomposite coated sponge structure with respect to applied pressure. It is not clear how formation of $\text{Fe}_2\text{O}_3/\text{C}@/\text{SnO}_2$ heterostructure helps in increasing the pressure range? Does growth of SnO_2 on Fe_2O_3 nanoneedles improve its mechanical properties?

Response:

Thanks very much for your valuable comments.

An n-n type heterostructure formed between Fe_2O_3 and SnO_2 promotes the transfer of electrons from Fe_2O_3 to SnO_2 , thus enhancing the conductivity of the pressure sensor. To prove that the formation of heterojunction in $\text{Fe}_2\text{O}_3/\text{C}@/\text{SnO}_2$ nanocomposites can improve the sensing performance of the $\text{Fe}_2\text{O}_3/\text{C}@/\text{SnO}_2$ pressure sensor, we measured the pressure response curves of the pressure sensors ($\text{Fe}_2\text{O}_3/\text{SnO}_2(3:4)$ and $\text{Fe}_2\text{O}_3/\text{C}@/\text{Fe}_2\text{O}_3(3:1:4)$). The response curves of the pressure sensors are presented in the supporting information of the revised paper (supplementary Fig. 5, page 4).

The sensitivity of $\text{Fe}_2\text{O}_3/\text{SnO}_2$ (3:4) sensor is $S_1 \sim 8.5 \text{ kPa}^{-1}$ when the pressure is below 10 kPa, $S_2 \sim 8.5 \text{ kPa}^{-1}$ when the pressure was ranged from 10 to 50kPa, and $S_3 \sim 8 \text{ kPa}^{-1}$ when the pressure was ranged from 50 to 150 kPa. It is apparently higher than that of single Fe_2O_3 pressure sensors ($S_1 \sim 3 \text{ kPa}^{-1}$, $S_2 \sim 3 \text{ kPa}^{-1}$ and $S_3 \sim 2 \text{ kPa}^{-1}$) and single SnO_2 pressure sensors ($S_1 \sim 1 \text{ kPa}^{-1}$,

$S_2 \sim 1 \text{ kPa}^{-1}$ and $S_3 \sim 0.6 \text{ kPa}^{-1}$). In addition, the sensitivity of $\text{Fe}_2\text{O}_3/\text{C}@\text{Fe}_2\text{O}_3$ (3:1:4) sensor is $S_1 \sim 70 \text{ kPa}^{-1}$ when the pressure is below 10 kPa, is $S_2 \sim 9 \text{ kPa}^{-1}$ when the pressure was ranged from 10 to 50 kPa, and $S_3 \sim 2 \text{ kPa}^{-1}$ the pressure was ranged from 50 to 150 kPa. The sensitivity of the $\text{Fe}_2\text{O}_3/\text{C}@\text{Fe}_2\text{O}_3$ (3:1:4) pressure sensor is lower than that of the $\text{Fe}_2\text{O}_3/\text{C}@\text{SnO}_2$ (3:1:4) pressure sensor ($S_1 \sim 680 \text{ kPa}^{-1}$, $S_2 \sim 98 \text{ kPa}^{-1}$ and $S_3 \sim 35 \text{ kPa}^{-1}$). We have characterized the SEM and TEM, which proved the formation of $\text{Fe}_2\text{O}_3/\text{SnO}_2$ heterojunction (Fig. 2g and supplementary Fig.2i). We also measured the SEM of the $\text{Fe}_2\text{O}_3/\text{C}@\text{Fe}_2\text{O}_3$ (3:1:4) as shown in below figures. It can be clearly seen that the morphology of $\text{Fe}_2\text{O}_3/\text{C}@\text{Fe}_2\text{O}_3$ (3:1:4) do not changed. The images of $\text{Fe}_2\text{O}_3/\text{C}@\text{Fe}_2\text{O}_3$ (3:1:4) reflecte a typical sea urchin-like structure. All results show that of $\text{Fe}_2\text{O}_3/\text{C}@\text{SnO}_2$ heterostructure can improve the sensing performance of pressure sensors.

We have added this description “To further prove that the formation of heterojunction in $\text{Fe}_2\text{O}_3/\text{C}@\text{SnO}_2$ nanocomposites can improve the sensing performance of the $\text{Fe}_2\text{O}_3/\text{C}@\text{SnO}_2$ pressure sensor. The sensing performance of the $\text{Fe}_2\text{O}_3/\text{SnO}_2$ (3:4) and $\text{Fe}_2\text{O}_3/\text{C}@\text{Fe}_2\text{O}_3$ (3:1:4) pressure sensors were measured (Supplementary Fig. 5 a-c). The sensitivity of $\text{Fe}_2\text{O}_3/\text{SnO}_2$ (3:4) sensor is $S_1 \sim 8.5 \text{ kPa}^{-1}$ when the pressure is below 10 kPa, $S_2 \sim 8.5 \text{ kPa}^{-1}$ when the pressure was ranged from 10 to 50 kPa, and $S_3 \sim 8 \text{ kPa}^{-1}$ when the pressure was ranged from 50 to 150 kPa. It is apparently higher than that of single Fe_2O_3 ($S_1 \sim 3 \text{ kPa}^{-1}$, $S_2 \sim 3 \text{ kPa}^{-1}$ and $S_3 \sim 2 \text{ kPa}^{-1}$) and single SnO_2 pressure sensors ($S_1 \sim 1 \text{ kPa}^{-1}$, $S_2 \sim 1 \text{ kPa}^{-1}$ and $S_3 \sim 0.6 \text{ kPa}^{-1}$). In addition, the sensitivity of $\text{Fe}_2\text{O}_3/\text{C}@\text{Fe}_2\text{O}_3$ (3:1:4) pressure sensor is $S_1 \sim 70 \text{ kPa}^{-1}$ when the pressure is below 10 kPa, $S_2 \sim 9 \text{ kPa}^{-1}$ when the pressure was ranged from 10 to 50 kPa, and $S_3 \sim 2 \text{ kPa}^{-1}$ when the pressure was ranged from 50 to 150 kPa (Supplementary Fig. 5c). The sensitivity of $\text{Fe}_2\text{O}_3/\text{C}@\text{Fe}_2\text{O}_3$ (3:1:4) pressure sensor is lower than that of $\text{Fe}_2\text{O}_3/\text{C}@\text{SnO}_2$ (3:1:4) pressure sensor ($S_1 \sim 680 \text{ kPa}^{-1}$, $S_2 \sim 98 \text{ kPa}^{-1}$ and $S_3 \sim 35 \text{ kPa}^{-1}$) (Supplementary Fig. 5b). The images of $\text{Fe}_2\text{O}_3/\text{C}@\text{Fe}_2\text{O}_3$ (3:1:4) reflect a typical sea urchin-like structure (Supplementary Fig. 6). All results show that of

Fe₂O₃/C@SnO₂ heterostructure can improve the sensing performance of pressure sensors.” in the revised paper, line 2, page 12.

4. Why Fe₂O₃/C response is constant at higher pressure range?

Response:

Thanks for your valuable comment.

The microfibers of sponge are composed of nanocomposites, and the increase of the contact leads to the increase of the current. With pressure increasing, the contact area of the sponge fiber reaches the maximum. Therefore, the conductive path reaches saturation. After, with the pressure increasing, the current change remains constant. The formula for calculating sensitivity is based on $S = (\Delta I / I_{\text{unloading}}) / \Delta p$. In addition, Fe₂O₃/C contains one structure (Fe₂O₃/C). Fe₂O₃/C@SnO₂ contains other structures (SnO₂@/C and Fe₂O₃/SnO₂ heterojunction). Under the same pressure, the current change of the Fe₂O₃/C@SnO₂ pressure sensor is higher than that of the Fe₂O₃/C pressure sensor. So, the Fe₂O₃/C response is constant at higher pressure range.

5. What are the dimensions of melamine sponge substrate used in the experiment? Does the dimension affect the pressure sensing range?

Response:

Thanks very much for your valuable comment.

“Melamine sponge was cut into a cuboid with a length of 19 mm, width of 19 mm and height of 4 mm.” (revised paper, line 22, page 20).

To analyze the dimensions of melamine sponge whether a factor influence the sensing performances of pressure sensors, we test the sensing performance of the Fe₂O₃/C@SnO₂ (3:1:4) pressure sensor with different dimensions. Firstly, performance of devices with different areas are shown in below figures. Melamine sponge was cut into a cuboid with a length of 15, 19, and 25 mm, width of 20, 19, and 25 mm, and height of 4 mm. Overall, the sensing performance of these devices are similar when the pressure changed from 0 to 150 kPa

The performance of devices with different thicknesses (2, 4, and 8mm) is shown in below figures. The variation of current ratio with the pressure of the sponges with difference thickness is similar, which indicates that the sensing performance of the sponge is independent of its thickness.

Overall, the sensitivity is similar of these devices with difference sponge dimensions. The pressure sensing range of these devices are different with the change of the sponge thicknesses. The compressibility of the sponge increases with its thickness. Therefore, we cannot rely on increasing the thickness of the sponge to increase the pressure response range of the device. Miniaturized devices are needed in many applications, such as wearable and human motions detection. In addition, the performance of our sensor is excellent compared with pressure responses reported in existing researches.

Material	Device size	Pressure range	Ref
Graphene @PU sponge	2 mm × 2 mm × 0.2 mm	0-10 kPa	1
CNT/rGO@PU sponge	13 mm × 8 mm × 5 mm	0-5.6 kPa	2
rGO/PANi@melamine Sponge	20 mm × 10 mm × 5 mm	0–27 kPa	3
graphene foam	2 mm × 7 mm × 7 mm	0-2 kPa	4
rGO foam	Diameter 45mm, thickness 2 mm	0-0.2 kPa	5
Fe ₂ O ₃ /C@SnO ₂	19 mm × 19 mm × 4 mm	0-150 kPa	in this work

We have added this description “The variation of current ratio with the pressure of the sponges with different areas and thicknesses is similar (Supplementary Fig.), which indicates that the sensing performance of the Fe₂O₃/C@SnO₂ sponge is independent of its area and thickness.

All results show that $\text{Fe}_2\text{O}_3/\text{C}@\text{SnO}_2$ pressure sensors are stable.” in the revised paper, line 1, page 14.

References:

1. Yao, H. B. et al. A Flexible and Highly Pressure-Sensitive Graphene–Polyurethane Sponge Based on Fractured Microstructure Design. *Adv. Mater.*, 25, 6692–6698 (2013).
2. Tiwari, A. et al. Highly Exfoliated Mwnt–Rgo Ink-Wrapped Polyurethane Foam for Piezoresistive Pressure Sensor Applications. *ACS Appl. Mater. Interfaces*, 10, 5185–5195 (2018).
3. Dong, X.C. et al. Flexible Pressure Sensor Based on Rgo/Polyaniline Wrapped Sponge with Tunable Sensitivity for Human Motions Detection. *Nanoscale*, 10, 10033 (2018).
4. Lv, L. et al. Ultrasensitive Pressure Sensor Based on an Ultralight Sparkling Graphene Block. *ACS Appl. Mater. Interfaces*, 9, 22885–22892 (2017).
5. Zang, X. et al. Unprecedented Sensitivity Towards Pressure Enabled by Graphene Foam. *Nanoscale*, 9, 19346–19352 (2017).

Response to referee 3’s comments:

1. The paper title is not appropriate because of the word “super working range”.

Response:

Thanks very much for your valuable comment.

“Sea urchin-like microstructures pressure sensors with ultra-sensitivity and super working range” have changed to “Sea urchin-like microstructures pressure sensor with ultra-broad range and high sensitivity”.

2. Under the headline of structural characterisation, the first para is not appropriate.

Response:

Thanks very much for your valuable comment.

The first paragraph has been moved below Fig. 1. (In page 4).

3. In the comparative study of graph 2 a, the first peak of composite ($\text{Fe}_2\text{O}_3/\text{C}@\text{SnO}_2$ 3:1:4) seems to be broadened instead of sharp. [In Page 6]

Response:

Thanks very much for your valuable comment.

We have changed “sharp” to “broad” (In page 6).

4. In page 7, 3rd line: “Either between or adjacent”. Both words should not be written in a statement.

Response:

Thanks very much for your valuable comment.

We have deleted “Either between or adjacent” in the revised manuscript.

5. In page 3: it’s mentioned that “carbon encloses the particles” and in page 7: it is mentioned that “carbon material was embedded in the gap” . Embedding and enclosing have different meaning.

Response:

Thanks very much for your valuable comment.

I’m sorry. The description of two parts was not comprehensive. We have modified two parts. “The acetylene black carbon encloses particles Fe_2O_3 , thereby forming $\text{Fe}_2\text{O}_3/\text{C}$ structure.” have changed to “One part of acetylene black carbon encloses particles Fe_2O_3 , whereas its other part of carbon materials was embedded in the gap Fe_2O_3 needles, forming $\text{Fe}_2\text{O}_3/\text{C}$ structure.” (In page 3) “a carbon material was embedded in the gap Fe_2O_3 needles, thus forming $\text{Fe}_2\text{O}_3/\text{C}$ structure.” have changed to “One part of acetylene black carbon encloses particles Fe_2O_3 , whereas its other part of carbon materials was embedded in the gap Fe_2O_3 needles, forming $\text{Fe}_2\text{O}_3/\text{C}$ structure.” (In page 6)

6. In page 8

- I. In Figure 3 there is no graph of carbon present.
- II. Graph 3 c and 3 d are same.
- III. Inside the graph “ev” written wrong. “v” should be capital.
- IV. In second last line 4844.9 eV has been written which should be 484.9

Response:

Thanks very much for your valuable comments.

- I. In Figure 3 there is no graph of carbon present.
We have added the graph of carbon. (Fig.3d, page 8).
- II. Graph 3 c and 3 d are same.
We have changed Fig. 3d to the XPS of carbon.
- III. Inside the graph “ev” written wrong. “v” should be capital.
We have changed “ev” to “eV”.
- IV. In second last line 4844.9 eV has been written which should be 484.9

We have changed “4844.9 eV” to “484.9 eV” (line 9, page 8).

7. In Fig 4 c, Caption: it should be detection of low pressure not weak pressure.

Response:

Thanks very much for your valuable comment.

We have changed “weak pressure” to “low pressure” (In page 9).

8. Page 10: 1st line: Of pressure sensor. “is” should be omitted.

Response:

Thanks very much for the valuable comment.

We have omitted the “is”.

9. Page 11: 1st para last line: The statement should not include obviously.

Response:

Thanks very much for the valuable comment.

We have omitted the “obviously”.

10. Page 12 : 1st para, Last 5th line from bottom: Either it should be “sensor is” or “sensors are” .

Response:

Thanks very much for the valuable comment.

It should be “sensor is”. We have changed “sensors is” to “sensor is” (line 11, page 13).

11. Page 13: Recovery time seems to be more than 22 ms. Why author have not considered the dip part (after 2000 ms) in the Fig. 4 d, before graph returns to its original state?

Response:

Thanks for the valuable comments.

The red circle in the figure is caused by the rapid rise of the experimental instrument.

Response time calculates the time of rising edge and falling edge in the references.

[Redacted]

Ref 1

Ref 2

Ref 3

References:

1. Chhetry, et al. Enhanced sensitivity of capacitive pressure and strain sensor based on $\text{CaCu}_3\text{Ti}_4\text{O}_{12}$ wrapped hybrid sponge for wearable applications. *Adv. Funct. Mater.* **30**,1910020, (2020).
2. Geun et al. Linearly and Highly Pressure-Sensitive Electronic Skin Based on a Bioinspired Hierarchical Structural Array. *Adv. Mater.* **28**, 5300-5306 (2016).
3. Bai, N. et al. Graded intrafillable architecture-based iontronic pressure sensor with ultra-broad-range high sensitivity. *Nat. Commun.* **11**, 1-9 (2020).
12. In section“High Pressure resolution”instead of using front wheel of a car author should use tensile testing equipment to show the sensor is having resolution in high pressure load also.

Response:

Thanks very much for your valuable comment.

The pressure sensor was subject to pressure values at 210 kPa as shown below. Then, the pressure sensor was added pressure of 2.8 and 25 kPa.

We have added this description “The sensitivity of the pressure sensor under high pressure was also test by the tensile test equipment. The sensing performance of device under the pressure of $P_0 = 210$ kPa is presented in Supplementary Fig. 9a-b. For the test, the device was first compressed to the reference pressure, followed by adding the pressure with two values of $\Delta P \sim 2.8$ kPa and $\Delta P \sim 25$ kPa, respectively.” in the revised paper, line 5, page 17.

This paper lack to show the innovative approach or novelty.

Response:

Thanks very much for your valuable comment.

This article is innovation as follows:

(1). Compared with previous work (as shown in table), the $\text{Fe}_2\text{O}_3/\text{C}@\text{SnO}_2$ pressure sensor exhibited high sensitivity (680 kPa^{-1}), broad range (up to 150 kPa) and good reproducibility (over 3500 cycles under a pressure of 110 kPa).

(2). We have found that the performance of pressure sensors can be improved when the heteostructure is used. And the heteostructure has rarely been employed in fabricating flexible pressure sensors. A depletion region and bands bending occurs in the contact sections of the heterojunction, which induces lower interfacial resistance and promoting of the charge transport/transfer. The heterostructure is formed between Fe_2O_3 and SnO_2 , which promotes the transfer of electrons from Fe_2O_3 to SnO_2 , thus enhancing the sensing performance of the pressure sensor.

(3) The sea urchin-like ternary nanocomposite $\text{Fe}_2\text{O}_3/\text{C}@\text{SnO}_2$ also contains $\text{Fe}_2\text{O}_3/\text{C}$ and $\text{SnO}_2@\text{C}$ hybrid structures. A ternary composite structure is rarely used in pressure sensors. The sea urchin-like Fe_2O_3 structure promotes signal transduction and protected Fe_2O_3 needles from mechanical breaking. Compared with single Metal oxide semiconductor, metal oxide semiconductor/conductor interface piezoresistive effect is essentially favorable for the change of contact area, which thus leads to big output current change. So, $\text{Fe}_2\text{O}_3/\text{C}$ and $\text{SnO}_2@\text{C}$ hybrid structures can improve the sensing performance of pressure sensor.

We also revised the second paragraph of the introduction section

The initial version was: There are two widely researched metal oxide including Fe_2O_3 and SnO_2 , because of their low-cost, environmental friendliness, and natural abundance. Studies have reported that coupling metal oxide and carbon compounds to form metal oxide/C nanocomposite, may improve photocatalytic²⁶ and electrochemical performances²⁷. Despite metal oxide/C nanocomposite processing large specific surface area and strong conductivity, they have rarely been employed in fabricating flexible pressure sensors.

In this work, we proposed a nanostructure design of materials with ultra-sensitivity for an ultra-broad-range pressure sensor. Particularly, this strategy involves the use of acetylene black

carbon as a carrier due to its strong conductivity and high specific surface. The acetylene black carbon encloses particles Fe_2O_3 , thereby forming $\text{Fe}_2\text{O}_3/\text{C}$ structure. Furthermore, one part of SnO_2 nanoparticles were dispersed into the carbon layer and formed $\text{SnO}_2@\text{C}$ structures, whereas its other part of nanoparticle adhered to the surface of Fe_2O_3 needles and formed $\text{Fe}_2\text{O}_3/\text{SnO}_2$ heterostructures. Carbon improves the conductivity of a single metal oxide. Collectively, the synergy of the three structures ($\text{Fe}_2\text{O}_3/\text{C}$, $\text{Fe}_2\text{O}_3/\text{SnO}_2$ and $\text{SnO}_2@\text{C}$) improved the limited pressure response range of a single structure. Notably, the $\text{Fe}_2\text{O}_3/\text{C}@\text{SnO}_2$ (3:1:4) pressure sensor exhibited high sensitivity (680 kPa^{-1}), fast response (10 ms), broad range (up to 150 kPa) and good reproducibility (over 3500 cycles under a pressure of 110 kPa).

The revision version was: High sensitivity can be obtained under two conditions: low initial current and large output current change under certain pressure²⁶. The conductivity of semiconductor is considerably low, so the initial current could achieve low level. In addition, semiconductor/conductor interface piezoresistive effect is essentially favorable for the change of contact area, which thus leads to big output current change²⁶. A depletion region and bands bending occurs in the contact sections of the heterojunction, which induces lower interfacial resistance and promotes of the charge transport/transfer²⁷. Therefore, heterojunction have been used in many modern devices, including light emitting diodes (LEDs), photodetectors and solar cells²⁸⁻³⁰. Therefore, when the metal oxide semiconductor/C composite structure and the heterostructure are used in the pressure sensor, the sensing performance of the pressure sensor may be improved.

In this work, we proposed a pressure sensor with nanostructure design of materials and the nanostructure contains metal oxide semiconductor/C composite structure and the heterostructure. The pressure sensor exhibited ultra-sensitivity and ultra-broad-range, when fabricated using the new nanostructure. We choose Fe_2O_3 and SnO_2 as sensing materials, because of their low-cost, environmental friendliness, and natural abundance. Sea urchin-like Fe_2O_3 is synthesized via a hydrothermal method. Particularly, this strategy involves the use of acetylene black carbon as a carrier due to its strong conductivity and high specific surface. The acetylene black carbon encloses particles Fe_2O_3 , thereby forming $\text{Fe}_2\text{O}_3/\text{C}$ structure. Furthermore, one part of SnO_2 nanoparticles were dispersed into the carbon layer and formed $\text{SnO}_2@\text{C}$ structures, whereas its other part of nanoparticle adhered to the surface of Fe_2O_3 needles and formed $\text{Fe}_2\text{O}_3/\text{SnO}_2$ heterostructures. Carbon improves the conductivity of a single metal oxide. Collectively, the synergy of the three structures ($\text{Fe}_2\text{O}_3/\text{C}$, $\text{Fe}_2\text{O}_3/\text{SnO}_2$ and $\text{SnO}_2@\text{C}$) improved the limited pressure response range of a single structure. Notably, the $\text{Fe}_2\text{O}_3/\text{C}@\text{SnO}_2$ (3:1:4) pressure sensor exhibited high sensitivity (680 kPa^{-1}), fast response (10 ms), broad range (up to 150 kPa) and good reproducibility (over 3500 cycles under a pressure of 110 kPa).

Materials	Nanostructure	Sensitivity (kPa ⁻¹)	Reference
ZnO/PDMS	Sea urchin-like	75 - 121 (0 - 200 Pa)	23 (Nat. Commun. 2018)
Au/Ag/PU	Sea urchin-like	2.46 (0 - 1 kPa) 0.52 (1 -8.2 kPa)	24 (Adv. Mater. 2016)
C/PDMS	Sea urchin-like	263 at 1 Pa	25 (Nat. Commun. 2020)
Mxene/Reduced Graphene Oxide Aerogel	Naosheets	0.55 (23 - 982 Pa) 3.81 (982Pa - 10 kPa) 2.52 (10 - 30 kPa)	36 (ACS Nano 2018)
Carbon Black@Polyurethane Sponge	Nanosheets	0.068 (0-2 kPa) 0.023 (2-10 kPa) 0.036 (10-16 kPa)	37 (Adv. Funct. Mater. 2018)
MXene/Sponge	Nanosheets	147 (0-5 kPa) 442 (5-20 kPa)	38 (Nano Energy .2018)
Graphene/Eco-flex	Triode-mimicking	4.68 (0-150 kPa) 11.09 (150-200 kPa)	39 (ACS Nano 2020)
CNT/cotton textile	Nanowires	14.4 (0-3.5 kPa) 7 (3.5-15 kPa)	40 (Adv. Mater. 2017)
MXene/PLA	Nanosheets	0.55 (0-3 kPa) 3.81 (3-10 kPa)	41 (Nano lett. 2019)
Fe ₂ O ₃ /C@SnO ₂ /Sponge	Sea urchin-like	680 (0-10 kPa) 98 (10-50 kPa) 35 (5-150 kPa)	In this work

REVIEWERS' COMMENTS

Reviewer #1 (Remarks to the Author):

The authors have revised the manuscript considerably with additional experiments. I do recommend the publication of the manuscript in the journal after minor revision.

Minor comment: English of the manuscript should be improved. For examples in the revised manuscript, grammatical mistakes such as "A depletion region and bands bending occurs", "heterojunction have", and "one part of SnO₂ nanoparticles were" are found.

Reviewer #3 (Remarks to the Author):

I have gone through your corrected paper and well satisfied by all the corrections have been made. But in the response of Query 12, the line should be "the pressure sensor was subjected" and the next line be "pressure sensor was added with pressure of 2.8 kPa", This is one minor change I would recommend and rest is fine.

Moreover, I am satisfied by your response of Query for explaining the innovative work in your research.

Your paper is ready to go for publication.

List of revisions made

1. According to referee 1's comment, English of the manuscript should be improved.
The English has been polished.
2. According to referee 3's comment, we have changed "For the test, the device was first compressed to the reference pressure, followed by adding the pressure with two values of $\Delta P \sim 2.8$ kPa and $\Delta P \sim 25$ kPa, respectively" to "During the test, the device was first subjected to the reference pressure. Then, the pressure sensor was added with pressure of 2.8 and 25 kPa, respectively."
3. According to journal requirements, we have modified the format of the manuscript.

Response to referees' comments

1. Minor comment: English of the manuscript should be improved. For examples in the revised manuscript, grammatical mistakes such as "A depletion region and bands bending occurs", "heterojunction have", and "one part of SnO₂ nanoparticles were" are found.

Response:

Thanks very much for your valuable comment.

The English has been polished. All changes are highlighted in yellow.

2. But in the response of Query 12, the line should be "the pressure sensor was subjected" and the next line be "pressure sensor was added with pressure of 2.8 kPa", This is one minor change I would recommend and rest is fine.

Response:

Thanks very much for your valuable comment.

We have changed "For the test, the device was first compressed to the reference pressure, followed by adding the pressure with two values of $\Delta P \sim 2.8$ kPa and $\Delta P \sim 25$ kPa, respectively" to "During the test, the device was first subjected to the reference pressure. Then, the pressure sensor was added with pressure of 2.8 and 25 kPa, respectively."